# Development and Application of a Stability Index Estimation Algorithm Based on Machine Learning for Elderly Balance Ability Diagnosis in Daily Life

**DOI:** 10.3390/bioengineering10080943

**Published:** 2023-08-08

**Authors:** Jeong-Woo Seo, Taehong Kim, Joong Il Kim, Youngjae Jeong, Kyoung-Mi Jang, Junggil Kim, Jun-Hyeong Do

**Affiliations:** 1Digital Health Research Division, Korea Institute of Oriental Medicine, Daejeon 34054, Republic of Korea; jwseo02@kiom.re.kr (J.-W.S.); jikim@kiom.re.kr (J.I.K.); dudwoj68@kiom.re.kr (Y.J.); jangkm@kiom.re.kr (K.-M.J.); 2Open XR Platform Convergence Research Center, Korea Institute of Science and Technology Information, Daejeon 34141, Republic of Korea; thkim@kiom.re.kr; 3KM Data Division, Korea Institute of Oriental Medicine, Daejeon 34054, Republic of Korea; 4Department of Biomedical Engineering, Konkuk University, Chungju 27478, Republic of Korea; junggill0219@gmail.com

**Keywords:** clinical test of sensory interactions with balance, stability index, balance ability diagnosis, force plate, machine learning

## Abstract

Background: The stability index estimation algorithm was derived and applied to develop and implement a balance ability diagnosis system that can be used in daily life. Methods: The system integrated an approach based on sensory function interaction, called the clinical test of sensory interaction with balance. A capacitance and resistance sensing type force mat was fabricated, and a stability index prediction algorithm was developed and applied using the center of pressure variables. The stability index prediction algorithm derived a center of pressure variable for 103 elderly people by Nintendo Wii Balance Board to predict the stability index of the balance system (Biodex SD), and the accuracy of this approach was confirmed. Results: As a result of testing with the test set, the linear regression model confirmed that the *r*-value ranged between 0.943 and 0.983. To confirm the similarity between the WBB and the flexible force mat, each measured center of pressure value was inputted and calculated in the developed regression model, and the result of the correlation coefficient validation confirmed an *r*-value of 0.96. Conclusion: The system developed in this study will be applicable to daily life in the home in the form of a floor mat.

## 1. Introduction

As population aging is accelerating worldwide, the demands for new proactive measures in social, economic, regional, and medical domains are increasing. Addressing social issues such as healthcare and welfare associated with the growing elderly population requires appropriate actions to extend lifespans and prevent the diseases and disorders commonly experienced by older individuals. Aging and disease often lead to reduced muscle strength and sensory function, causing a decline in physical motor and cognitive abilities among the elderly. Notably, a diminished lower extremity muscle strength, vestibular function, and visual function can contribute to falls and abnormal motor function.

Recently, various studies focusing on elderly diagnosis have been conducted on various applications of machine learning or artificial intelligence in research subjects and health sciences for postural control and various evaluation methods [1,2,3,4,5]. These studies derive experiment-based diagnostic algorithms and confirm their effectiveness. These algorithms can be applied to the developed measurement system and applied to medical devices or healthcare devices that can be used in clinical or home settings. In particular, a reduced balance ability, i.e., a weakened capacity to maintain body position by keeping the center of gravity within specific boundaries, increases the risk of falls owing to an intricate interplay between the sensory and musculoskeletal systems [6].

Numerous methods have been developed to evaluate balance ability characteristics, particularly through dynamic functional assessment methods targeted at older adults. One such method is the one-leg standing (OLS) test, which measures the duration an individual can stand on one lower limb without support. The OLS test has been employed in diagnosing the musculoskeletal ambulation disability symptom complex [7] and serves as a practical tool for frailty screening in the elderly. Another assessment technique is the timed-up and go (TUG) test, which involves tasks such as sit-to-stand, walking a distance of 3 m, turning, gait analysis, and returning to a seated position. Previous studies have established a correlation between the gait kinematic variables measured during the TUG test and the Community Balance and Mobility Scale (CBMS) [8]. Dynamic functional evaluations like OLS and TUG are based on comprehensive perspectives, such as lower extremity muscle strength, rather than on sensory functions (e.g., vision and vestibular function) that directly affect balance ability. However, balance ability is a function of interaction between detailed sensory functions; hence, detailed methods to quantify balance ability are needed.

One such evaluation tool is the Clinical Test of Sensory Interaction with Balance (CTSIB), which has been standardized to measure the sense of balance and behaviorally assess the processing of vestibular and proprioceptive senses [9]. Traditionally, the CTSIB records the duration of maintaining a standing posture under various vision conditions, vestibular sensory interferences, and proprioceptive sensory interferences. Recently, quantitative measurement methods considering the movement range have incorporated the use of the center of mass (CoM) obtained through inertia measurement units or the center of pressure (CoP) derived from force plates. An example is the instrumented CTSIB (i-CTSIB), which has been used in clinical practice to measure CoP and calculate evaluation variables while maintaining standing balance. This method has been implemented through several commercial devices, such as the BODITRAK BALANCE Sensor [10], which generates a pressure map employing flexible sensors that are portable and adaptable to any surface. Another system, BTrackS™ (Balance Tracking Systems, Inc., San Diego, CA, USA) [11], employs a highly accurate force sensor enabling the evaluation of various aspects of balance ability. However, these commercial products typically require measurements to be conducted following procedures suggested by clinicians or professional assessors, thereby limiting their applicability for home visits or non-clinical settings.

With the increasing elderly population, the capacity of the existing medical system to provide comprehensive care is approaching its limit [12]. Moreover, current evaluation methods often require direct visits to hospitals, leading to delayed assessments when balance ability has already significantly deteriorated. Consequently, there is a pressing need for a system that allows for easy and regular measurement of balance ability in everyday life [13]. There are also more accessible systems, such as the Nintendo Wii balance board, that offer a precise assessment of balance ability [14,15]. Such systems apply a balance assessment algorithm and can be easily used at home. However, they also have the disadvantage of needing to be set up and connected for each measurement. Commercialized force plate products have the disadvantage of requiring the installation of a measuring device for each measurement, and there is a requirement for a measurement system that can be installed in the form of a floor mat in the home and used for general purposes in daily life. Thus, measuring equipment that can be used at home and a diagnostic algorithm that can guarantee measurement accuracy are required. Furthermore, such a system should be developed towards virtual or augmented reality in a non-face-to-face environment in the future so that it can be used for telemedicine diagnosis and treatment in a state where immersion is increased.

In this study, the stability index (SI) calculated from CoP was measured when performing CTSIB in the commercially available balance ability diagnosis system. This was defined as an individual balance ability score, and a machine-learning-based balance ability evaluation algorithm was developed using various variables calculated from the CoP measured when performing CTSIB on a load-cell-based force plate Nintendo Wii balance board. In general, the SI is simply calculated as the amount of movement of the AP and ML axes [16]. Therefore, more specifically, we develop a model to determine which CoP variables affect the SI prediction algorithm. A hybrid sensing (capacitance and resistance) type ground reaction force measurement mat was developed, and an SI estimation algorithm was applied. Following this, its usability was confirmed through verification.

This is aimed at identifying more detailed balance ability characteristics. According to the identified characteristics, a balance ability diagnosis algorithm is developed. The purpose of this derived system and algorithm is to make it possible to apply it to a virtual reality (VR) or augmented reality (AR) environment system with good immersion and accessibility in the future.

## 2. Materials and Methods

### 2.1. Study Procedure

In this study, to develop a balance diagnosis algorithm, the SI was measured in a commercially available system for elderly people. To identify correlated variables and develop an algorithm for estimating the SI, CoP measurements and variable calculations were performed while performing CTSIB on a commercially available force plate for the same subject. A regression equation for estimating the SI was derived through various machine learning, and validation was performed by applying this regression equation model to the developed flexible force mat. And, it verified that the developed SI algorithm can be applied to the developed force mat (Figure 1).

### 2.2. Development and Validation of the Balance Ability Model

A total of 103 elderly individuals participated in the development of the balance ability evaluation model, comprising 27 males: age (79.78 ± 4.76 years), height (162.72 ± 6.11 cm), weight (67.84 ± 9.82 cm); and 81 females: age (77.73 ± 5.21 years), height (147.37 ± 3.31 cm), weight (54.03 ± 6.42 cm). First, a sufficient explanation of the experiment was given to the subjects, and informed consent was obtained. The sample size is 85 if population size is 108. This means that 85 or more measurements/surveys are needed to achieve a confidence level of 95% that the real value is within ±5% of the measured value. The data acquisition procedure consists of SI measurement in the balance system, CTSIB CoP measurement in the Nintendo Wii balance board (WBB, Nintendo Inc., Kyoto, Japan), and development and verification of the SI prediction model using measurement data. Second, for system and algorithm validation, 30 new subjects were recruited, and the values obtained by overlapping the WBB and the flexible force mat were applied to the developed model and the SI values derived were compared. Initially, the SI of each of seven CTSIB conditions (eye open/close with firm, one-foam, two-foam, and eye open-down count with firm (50 to 0, subtract by 1)) were measured using the balance system (Biodex SD) [16]. Subsequently, 30 s of data per CTSIB condition were collected using the commercially available force plate WBB, and the CoP values were calculated. The datasets were processed for the seven conditions across the 76 participants. To mitigate the high-frequency and low-frequency noise signals in the x- and y-axis components of the measured CoP data, trend factors were analyzed. Butterworth fourth order bandpass filtering (cut-off frequency: 0.1 Hz to 4 Hz) was then applied to the noise-removed CoP data [14]. Sway variables, which would serve as inputs for the machine-learning model, were derived from the filtered CoP data (Figure 2).

Table 1 presents the 10 variables calculated from the time domain data, including the mean, mean path, mean distance, mean acceleration, ellipse area, high 5%, low 5%, center, and median frequency.

For the training and evaluation of the SI estimation, the total dataset was divided into a 70% training set and a 30% test set. Four regression analyses were conducted: linear regression, support vector machine (SVM) regression, generalized additive model (GAM) regression, and tree regression. The training process was conducted using randomly selected training sets, and the model’s performance was assessed using the test set [17]. During the training process, Bayesian optimization was employed to determine the hyper-parameters for each of the four models. The process involved conducting 30 iterations of repeated learning without any imposed time constraints. The acquisition function utilized in this study was set as the probability of improvement (POI). The chosen linear regression model incorporated L2 regularization as a hyper-parameter, which was determined through an optimization procedure. The CoP variables measured when performing i-CTSIB on the flexible force mat were used as an input value (x) for the model developed with a 70% training set on the 30% of the subjects left as a test set. The correlation coefficient (*r*-value) between the observed and predicted values of the response variable was computed to evaluate the model’s accuracy.

### 2.3. Customized CoP Measurement System

The sensing mechanism of the pressure sensor is designed to detect changes in capacitance and resistance values simultaneously. Each sensor cell measured 21 mm × 36 mm × 2.5 mm, and a total of 3264 cells were arranged in a grid pattern measuring 68 × 48, resulting in a total size of 1530 mm × 1820 mm (Figure 3).

A. To enhance safety and prevent slippage when used by elderly individuals, a 15 mm thick safety mat made of PE material was affixed to the upper plate of the force mat. Power was supplied using DC 5.0 V, 2.0 A via a Micro USB connection. The data required for measurement were obtained through a dedicated PCB, with a sampling frequency of 23 Hz. The force range of the mat spanned from 10 kg to 100 kg. Communication between the force mat and the console PC was achieved using the USB-connected serial port communication method. The pressure distribution measured by the flexible force mat, when connected to the console PC, was visualized as an image representing the pressure distribution.

### 2.4. System and Algorithm Validation

To assess the similarity between the flexible force mat and the commercially available force plate WBB, a comparative verification experiment was conducted with 30 new participants (7 males and 23 females). This group comprised 7 males: age (76.34 ± 3.87 years), height (166.21 ± 4.02 cm), weight (66.72 ± 7.14 cm); and 23 females: age (78.14 ± 4.20 years), height (150.16 ± 3.31 cm), weight (54.03 ± 6.42 cm). The flexible force mat was positioned on the commercially available WBB, and CoP measurements were simultaneously collected. Using the same algorithm, variables were calculated based on the collected data. The SI was then derived by inputting the calculated variables into the linear regression model that was ultimately selected. The similarity between the two devices was evaluated by conducting a correlation analysis (Pearson’s r) between the SI derived from the measurements obtained by both devices.

## 3. Results

### 3.1. Results of CoP Variables during CTSIB

Table 2 presents the CoP variables measured during the performance of CTSIB on a commercially available force plate, WBB. The results indicate that variables such as the mean path, velocity, and distance showed an increasing trend when participants performed the task with their eyes closed and as the difficulty level of the standing posture increased.

### 3.2. Results of Stability Index Regression Model

The correlation coefficient (*r*-value) was calculated to determine the relationship between the observed SI and the predicted SI generated by the regression model. A regression model was developed using the training set, employing four types of regression models: linear regression, SVM, GAM, and regression tree. Table 3 shows the results of validating the model with the test set. All four regression models demonstrated a correlation coefficient (*r*-value) of 0.7 or higher, with a majority achieving high levels of accuracy at 0.8 and 0.9. Among these models, the linear regression model exhibited the best performance, displaying *r*-values ranging from 0.943 to 0.983. An example graph of the real and predicted values of the test set is shown in Figure 4.

To formulate the regression model that yielded the highest performance among the different regression models, beta and bias values for each CTSIB task condition were extracted and incorporated into the final formula (Figure 5).

### 3.3. Results of System and Algorithm Validation

To assess the similarity and accuracy between the WBB and the flexible force mat, a correlation analysis (Pearson’s r) was conducted to compare the SI values derived from the measured values. The analysis revealed a high correlation coefficient of 0.96 (Pearson’s r, *p* < 0.05, α = 0.05).

## 4. Discussion

To validate the similarity between the load-cell-based WBB force plate and the newly developed force mat, we evaluated the SI estimation model results that calculated the CoP data simultaneously measured by both systems. The results demonstrated a high accuracy, with a correlation coefficient (*r*-value) of 0.96 between the values calculated by the linear regression model used to estimate the SI. This indicates that, despite potential differences in the unit and resolution of the measured force values, there is no statistically significant distinction in calculating the CoP, which is essential for accurate balance evaluation [18,19]. Consequently, the reliability of the CTSIB-based balance ability evaluation using the force mat can be considered assured.

Previous studies have explored various endeavors to produce an inexpensive, portable, and flexible sensor-based force plate, akin to the system developed in our study, with the aim of utilizing it as a commercial force plate equivalent. Notably, these studies have demonstrated promising outcomes. For instance, one study employed a textile force mat system that wirelessly transmits pressure to confirm that there was no statistical difference in foot pressure compared to a commercial platform during walking and one-leg standing [20]. Another study investigated the feasibility of measuring the ground reaction force by utilizing a graphene-based flexible sensor and acquiring walking data with a pressure sensor [21]. Although the shape and material composition of the force sensors differ, the findings indicate the potential for measuring walking and standing posture [22,23]. The flexible nature of the sensor, allowing for folding or rolling, provides a distinct advantage over conventional thick and heavy force plates. This advantageous feature holds significant value in the implementation of XR system interfaces, offering enhanced accessibility at home, ease of setup, and cost-effectiveness. Moreover, an additional benefit lies in the capability of assessing balance ability using the XR system within the comfort of one’s own home [24].

When it comes to sensor-based hardware, variations in materials and development methods may exist, but there are no discrepancies in the measured force or pressure values. Therefore, the crucial focus lies in developing a robust balance ability evaluation algorithm using such a system and ensuring its accuracy. Numerous methods are available for assessing balance ability, and among them, the CTSIB evaluates balance performance based on sensory interaction and the time allocated for performing tasks under different sensory conditions. In recent years, research has progressed towards employing instrumented tests that quantitatively evaluate balance using the center of pressure (CoP) variables measured and calculated with a force plate during CTSIB assessments [25,26,27]. For instance, a study examined the stance postural control of ambulant children with cerebral palsy, utilizing CoP-derived variables such as velocity, range, and area to compare them with typically developing children [28]. Another study investigated the variations in CoP variables according to age group and different ground conditions during CTSIB [29]. These examples illustrate the potential for quantification, enabling independent evaluations and quantified scoring even without direct involvement from a balance tester.

To establish a mathematical model for balance evaluation based on the SI, CoP variables were measured and calculated for elderly individuals. The advantage of SI is that it can confirm absolute and objective results in the evaluation of balance ability [30]. Since it is a score that quantifies the amount of movement starting from zero and proportions, it is easy to determine the level of change in balance ability even in repeated measurements over time [31]. In addition, the Berg balance scale, which is known as a task used for balance ability, is highly dependent on the measurer because the subjective opinion of the clinician based on the questionnaire is involved [32]. If the follow-up of the balance ability using the device is performed, it can be used as an indicator of a more objective result; in this study, it is advantageous to predict the SI indicator by using the correlation of various CoP-related variables without using expensive equipment to directly calculate these indicators.

Various techniques were employed to develop regression models using these variables and, in most cases, a high accuracy was achieved with correlation coefficients (*r*-values) of 0.7 or higher. Notably, both the linear regression model and the regression tree model consistently demonstrated a high accuracy, exceeding 0.9 in all cases. The regression tree model functions by estimating the result value through averaging at the endpoint nodes, employing a predictive approach that creates splits based on predictors that minimize the sum of squared errors. In our study, considering the similar accuracy results obtained, we ultimately selected the linear regression model. This choice allows for the derivation of a simple equation-based formula and facilitates the easier development and utilization of the algorithm.

Software programming was performed to establish the necessary connections between the hardware force mat and the developed algorithm, as well as to create a user-friendly graphical user interface (GUI) for presenting the resulting values. During the performance of the seven CTSIB tasks, CoP values were sequentially measured and stored. After completing the final measurement, these values were input into the SI calculation algorithm, and the results were generated. The GUI consists of two main components: a visualization of the measured CoP trajectory and a display of the calculated and reported result values. The scores for each of the seven tasks allow for differentiation based on visual conditions (on/off) and task difficulty, reflecting the unique characteristics of each task (Figure 6). The overall SI is presented as the mean value, and the balance ability assessment is designed to enable the monitoring of score trends over time in connection with future balance training programs.

The ultimate goal of developing this balance ability assessment system is to incorporate features such as visual tilting to introduce vestibular interference and the option to block visual input using VR head-mounted displays (HMDs). Examining the application of XR systems, including the force mat developed in this study, the Virtual Reality Floor Mat Activity Region is an XR system that utilizes force plate technology and integrates with HMDs [33]. This setup provides a fully immersive experience, addressing real-world challenges faced by many VR users, while also ensuring safer gameplay in domestic environments. Similarly, the system developed in this study can be applied to provide immersive balance ability assessments and implement avatars in remote locations [34]. Furthermore, due to its large-area force mat design, this system can facilitate dynamic motor function evaluations such as walking and timed up and go tests.

One limitation of this study is that it was not based on large-scale data, necessitating an optimization process for the algorithm. Addressing this limitation will involve accumulating a substantial amount of data and more advanced deep learning techniques for future analysis and improvement.

In this study, we developed a CTSIB-based balance ability evaluation system that can be conveniently conducted at home, outside of a clinical setting. This system was designed as an integral component of the XR environment with the potential for scalability. By implementing an XR system composed of a force mat, it becomes feasible to perform i-CTSIB measurements within head-mounted displays such as virtual reality and augmented reality, allowing for conditions such as visual-off and enabling an interaction-based measurement system for remote medical diagnosis. In future research, we plan to apply it to XR HMD to confirm its usability.

## 5. Conclusions

A linear regression model capable of estimating the feature SI was developed based on CTSIB CoP measurement data of elderly subjects to evaluate the balance ability when performing CTSIB; as a result of testing with the test set, the *r*-values of seven CTSIB tasks were obtained with values ranging from 0.943 to 0.983, confirming a high prediction accuracy. The SI prediction can be used instead of expensive commercialized systems, and a model that can predict CoP variables that affect balance ability was identified and developed. This model is more meaningful because it is based on measurements of real elderly people. The system developed in this study will be applicable in daily life in the home in the form of a mat.

## Figures and Tables

**Figure 1 bioengineering-10-00943-f001:**
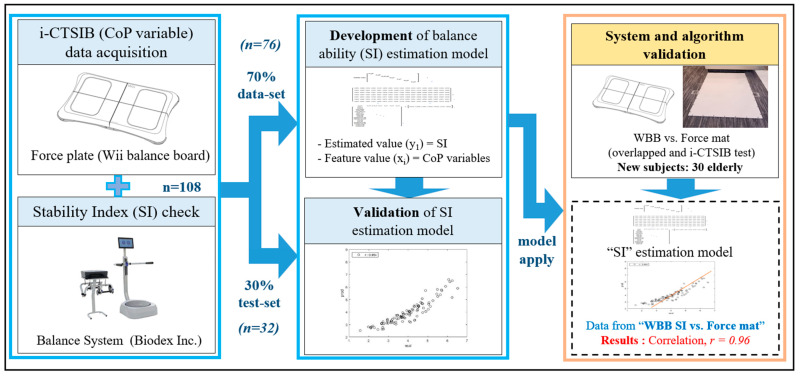
Study procedure of data acquisition, model development, and validation of system.

**Figure 2 bioengineering-10-00943-f002:**
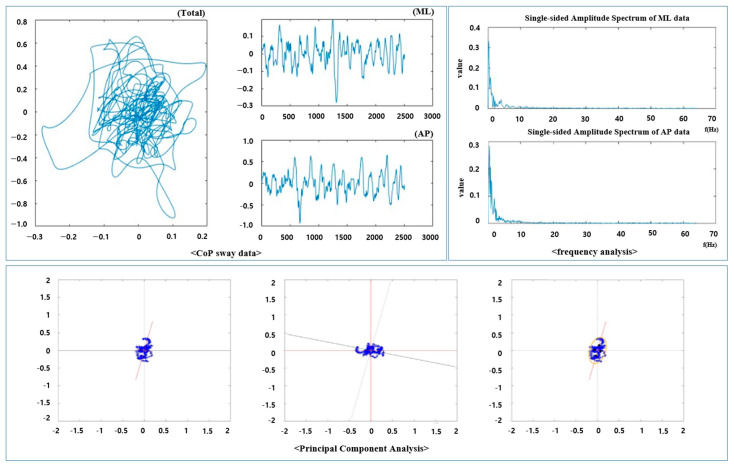
Analysis of CoP feature measured by commercialized force plate.

**Figure 3 bioengineering-10-00943-f003:**
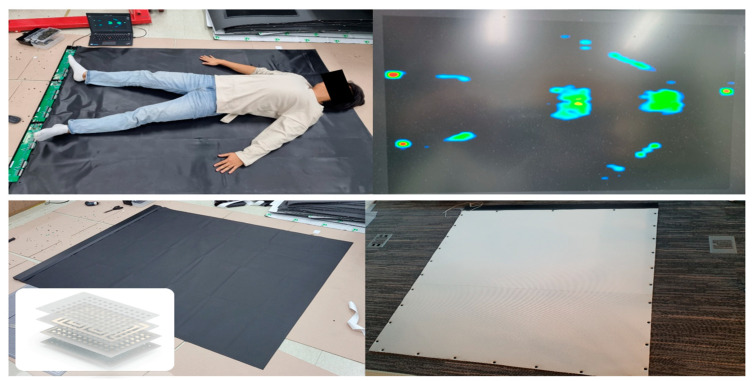
Flexible force mat.

**Figure 4 bioengineering-10-00943-f004:**
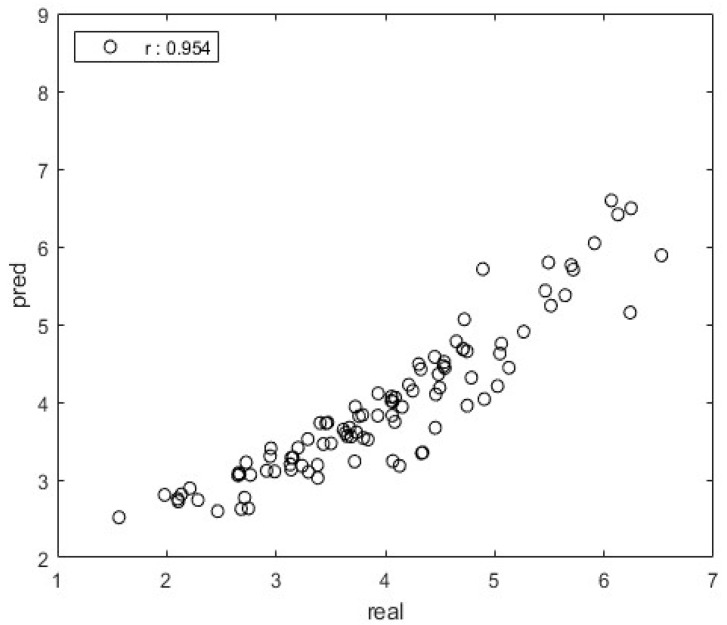
Example of validation of linear regression model (example: eye-open, firm condition).

**Figure 5 bioengineering-10-00943-f005:**
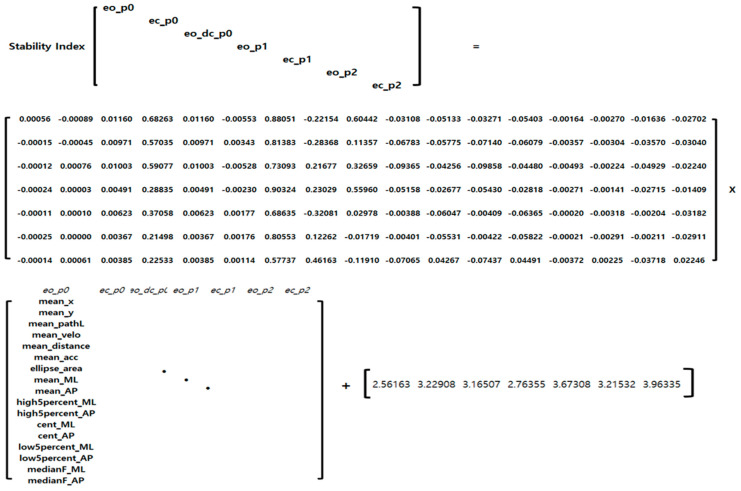
Regression model for estimating the stability index.

**Figure 6 bioengineering-10-00943-f006:**
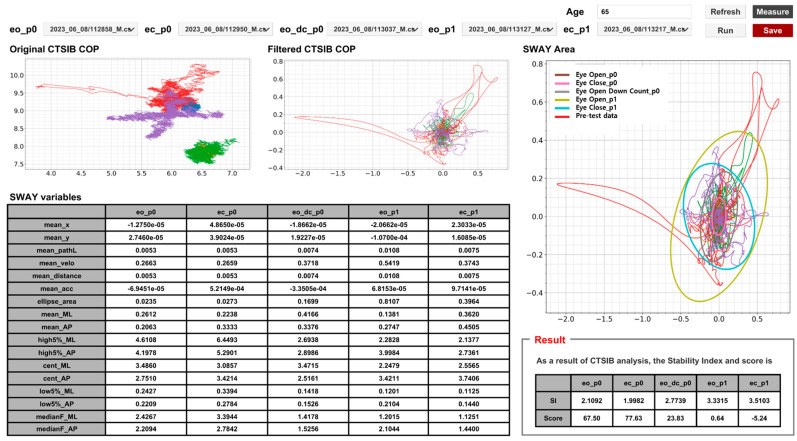
Example of results GUI according to the i-CTSIB.

**Table 1 bioengineering-10-00943-t001:** Description of CoP variables.

Variables	Description
mean [cm]	mean of CoP movement
mean path [cm/s]	mean of CoP distance per second
mean velocity [cm/s]	mean of CoP velocity per second
mean distance [cm]	mean of one sample distance per second
mean acc. [cm/s^2^]	mean of CoP acceleration per second
ellipse area [cm^2^]	95% area based on ellipse major axis
frequency high 5% [Hz]	top 5% power frequency of CoP
frequency center [Hz]	center frequency of CoP
frequency low 5% [Hz]	bottom 5% power frequency of CoP
frequency median [Hz]	median frequency of CoP

CoP: center of pressure.

**Table 2 bioengineering-10-00943-t002:** Results of CoP variables during CTSIB.

Variable	Eye-Open				Eye-Close		
	Firm	Firm-DC	Foam-1	Foam-2	Firm	Foam-1	Foam-2
mean [cm]	0.39 ± 0.09	0.38 ± 0.09	0.38 ± 0.08	0.39 ± 0.09	0.38 ± 0.09	0.38 ± 0.07	0.38 ± 0.08
mean path [cm/s]	0.02 ± 0.01	0.03 ± 0.02	0.03 ± 0.01	0.04 ± 0.01	0.03 ± 0.02	0.04 ± 0.02	0.05 ± 0.02
mean velocity [cm/s]	1.20 ± 0.54	1.69 ± 0.97	1.59 ± 0.56	2.04 ± 0.65	1.64 ± 0.94	2.26 ± 1.05	2.94 ± 1.02
mean distance [cm]	0.02 ± 0.01	0.03 ± 0.02	0.02 ± 0.01	0.04 ± 0.01	0.03 ± 0.02	0.04 ± 0.02	0.05 ± 0.02
mean acc. ×10^3^ [cm/s^2^]	−0.16 ± 0.78	−0.12 ± 1.19	−0.20 ± 0.67	−0.08 ± 1.51	−0.08 ± 0.76	−0.10 ± 1.15	−0.03 ± 1.23
ellipse area [cm^2^]	1.49 ± 1.67	3.27 ± 5.73	2.13 ± 1.50	3.58 ± 2.24	1.71 ± 1.79	3.16 ± 3.00	6.28 ± 4.07
frequency high 5% [Hz]	2.73 ± 0.76	2.73 ± 0.72	2.73 ± 0.63	2.77 ± 0.69	2.82 ± 0.66	2.84 ± 0.64	2.78 ± 0.62
frequency center [Hz]	2.87 ± 0.80	2.87 ± 0.76	2.87 ± 0.67	2.92 ± 0.73	2.97 ± 0.70	2.99 ± 0.67	2.93 ± 0.66
frequency low 5% [Hz]	0.14 ± 0.04	0.14 ± 0.04	0.14 ± 0.03	0.15 ± 0.04	0.14 ± 0.03	0.15 ± 0.03	0.15 ± 0.03
frequency median [Hz]	1.44 ± 0.40	1.44 ± 0.38	1.44 ± 0.33	1.46 ± 0.37	1.48 ± 0.35	1.49 ± 0.34	1.46 ± 0.33

**Table 3 bioengineering-10-00943-t003:** The test set r-score results of regression models during CTSIB.

Regression Model	Eye-Open				Eye-Close		
	Firm	Firm-DC	Foam-1	Foam-2	Firm	Foam-1	Foam-2
Linear	0.954	0.943	0.956	0.983	0.945	0.961	0.970
SVM	0.898	0.905	0.905	0.968	0.891	0.913	0.938
GAM	0.865	0.786	0.789	0.793	0.805	0.821	0.862
Tree	0.954	0.943	0.956	0.983	0.945	0.961	0.970

Linear: linear regression, SVM: support vector machine regression, GAM: generalized additive model regression, tree: tree regression.

## Data Availability

The data presented in this study are available on request from the corresponding author.

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
