# Peer review of "Development and Application of a Stability Index Estimation Algorithm Based on Machine Learning for Elderly Balance Ability Diagnosis in Daily Life"

_bioengineering, 2023, doi:10.3390/bioengineering10080943_

Round 1

Reviewer 1 Report

Dear authors, 

Some comments to improve. 

1.       The introduction could be expanded to better contextualize the need for a balance ability diagnosis system for elderly people. It would be helpful to include relevant statistics on the prevalence of falls and balance-related injuries in this population, and to briefly summarize the current state of the art in balance assessment tools. Please check latest articles on Sensors (MDPI) including https://pubmed.ncbi.nlm.nih.gov/29985149/ and https://pubmed.ncbi.nlm.nih.gov/25268919/ and https://pubmed.ncbi.nlm.nih.gov/36433479/

Note: I am not an author of any of the above.

2.       The methods section could be more detailed to allow for replication of the study. Specifically, more information is needed on the fabrication of the piezo-resistive force mat, the selection of participants, and the data collection procedures.

Sample size is need calculation.

Lines 165-167 I don’t understand how these 30 participants were used to validate? I got confused there. System and Algorithm Validation need more details. How is the characteristics of these. What is meant by new? Not included in main?

3.       The results section could be better structured to clearly present the findings. It is unclear from the current text how the Stability Index prediction algorithm was derived and applied using the CoP variables, and how the accuracy of the algorithm was confirmed through the correlation between real and predicted values. Results of CoP variables during CTSIB better be Figure.

4.       The discussion section could be expanded to provide more context for the results and to discuss their implications for future research and practice. For example, what are the potential advantages and limitations of using the Stability Index estimation algorithm in clinical settings, and how does it compare to other balance assessment tools?

5.       The conclusion section could be more concise and focused on the main takeaways from the study. It would be helpful to summarize the key findings and their potential impact on the field of balance ability diagnosis for elderly people. Table 3. The test-set R-score results of regression models during CTSIB show that firm is lower should it not be higher?

6.       References need to add DOI. Figures need better resolution. 

Author Response

Response to Reviewer 1 Comments

Dear Reviewer 1.

We sincerely responded to the opinions you gave us and implemented a revision of the manuscipt.

Thanks to your meaningful comments, we were able to improve the quality of this manuscript. Thanks again for the review.

Sincerely,

 From Corresponding author.

 Point 1: The introduction could be expanded to better contextualize the need for a balance ability diagnosis system for elderly people. It would be helpful to include relevant statistics on the prevalence of falls and balance-related injuries in this population, and to briefly summarize the current state of the art in balance assessment tools. Please check latest articles on Sensors (MDPI) including https://pubmed.ncbi.nlm.nih.gov/29985149/ and https://pubmed.ncbi.nlm.nih.gov/25268919/ and https://pubmed.ncbi.nlm.nih.gov/36433479/

Response 1: (line 87-90) The papers you suggested are studies that confirm the feasibility of using the Wii balance board in the home. We have presented the advantages of the papers, such as the possibility of using it at home, and the disadvantages, such as the setting. Thank you for your good comments. The introduction has been supplemented.

Point 2: The methods section could be more detailed to allow for replication of the study. Specifically, more information is needed on the fabrication of the piezo-resistive force mat, the selection of participants, and the data collection procedures. Sample size is need calculation.

Response 2: (lines 132~144, line 136~138) To complement the information on this mat, a sensing type (hybrid: detect changes in capacitance and resistance values) was added, and the internal configuration diagram of the sensor was supplemented in “figure 3”. In addition to age, height and weight information were added to the experiment participant information. In addition, “Figure 1” was modified to supplement the data collection process, and the contents of the process were written in “lines 132 to 144”. The sample size is 85. This means 85 or more measurements/surveys are needed to have a confidence level of 95% that the real value is within ±5% of the measured/surveyed value (line 136~138). Thank you.

Point 3: Lines 165-167 I don’t understand how these 30 participants were used to validate? I got confused there. System and Algorithm Validation need more details. How is the characteristics of these. What is meant by new? Not included in main?

Response 3: (Figure 1) There was an error in the description. In this study, 108 subjects were used to create the S.I estimation model, of which 76 (70%) were used as the training-set and 32 (30%) as the test-set, and the results are shown in Table 3. 30 new paticipants is an additional experiment for system and algorithm validation. This is to check the correlation between the S.I results derived by inputting the values (CoP variables) ​​measured by overlapping the WBB and Force mat into the algorithm. The figure 1”is written to compensate for the error in the explanation. thank you.

Point 4: The results section could be better structured to clearly present the findings. It is unclear from the current text how the Stability Index prediction algorithm was derived and applied using the CoP variables, and how the accuracy of the algorithm was confirmed through the correlation between real and predicted values. Results of CoP variables during CTSIB better be Figure.

Response 4: (line 131~144) I think it is correct to explain how the Stability Index prediction algorithm was derived and applied using the CoP variables in more detail in the method “2.2. Development and validation of balance ability model”. It can be confirmed with the supplementary information in the Development and validation of balance ability model section. Please understand that the results of CTSIB are not presented in figure form because each variable is derived as a non-linear value.

Point 5: The discussion section could be expanded to provide more context for the results and to discuss their implications for future research and practice. For example, what are the potential advantages and limitations of using the Stability Index estimation algorithm in clinical settings, and how does it compare to other balance assessment tools?

Response 5: (lines 291~301) The target value for prediction in this study, S.I, is clinically meaningful because it is an index that can be followed up more objectively and quantitatively. In contrast, assessments such as the Berg balance scale tend to be highly questionnaire-based, highly dependent on the measurer, which has a subjective disadvantage. These contents have been written and added to the discussion lines 291~301.

Point 6: The conclusion section could be more concise and focused on the main takeaways from the study. It would be helpful to summarize the key findings and their potential impact on the field of balance ability diagnosis for elderly people. Table 3. The test-set R-score results of regression models during CTSIB show that firm is lower should it not be higher?

Response 6: The balance ability estimation model and data acquisition results similarity were confirmed that between the WBB and flexible force mat. The S.I prediction can be used instead of expensive commercialized systems, and a model that can predict CoP variables that affect balance ability was identified and developed. This model is more meaningful because it is based on measurements of real elderly people. Added this to the conclusion. And I agree that the R-score of the firm model should be higher. However, as the difficulty level increases according to the installation of foam, there is no individual difference in difficulty between subjects, so I think that the accuracy of the predictive model will be higher. Confirmation of this will be presented through a more detailed analysis in a later study. Thank you for your understanding.

Point 7: References need to add DOI. Figures need better resolution. 

Response 7: Based on your opinion, we have added all DOIs of the reference. thank you.

Reviewer 2 Report

Dear Authors,

First of all, I would like to congratulate you on the effort you have put into this research. The study presented is of interest to the community of computer engineers and healthcare professionals who carry out their care activities in relation to elderly people with a high risk of falling. However, the manuscript has formal errors and methodological limitations that need to be addressed before publication in this Journal.

ABSTRACT:
Abbreviations are discouraged in this section. Please remove them.

INTRODUCTION:
Although this is a fairly extensive section, the small number of bibliographical references used by the Authors to set out the theoretical background of the subject matter is striking. There is a lack of recent and/or relevant references on the object of study of postural control and its different modes of evaluation (doi: 10.3390/su12031222) or on the different applications of artificial intelligence in Health Sciences.

METHODS:
This section should begin by defining the research methodology that has been applied. In this regard, authors should use the relevant EQUATOR Network checklist to ensure that they provide all the necessary information (and in the right order) throughout the manuscript (but especially in this section to facilitate the reproducibility of the research).

RESULTS:

Tables should make all abbreviations explicit at the bottom.
Descriptive results such as mean or standard deviation should include only one decimal place.

DISCUSSION:
This section suffers from the same shortcomings as the Introduction in relation to the limited use of bibliographical references to support the Authors' arguments. Right now, many sentences, lacking references, seem like speculation.

CONCLUSIONS:
The clinical impact of the research is not adequately reflected in this section.

Kind regards

I have detected grammatical and syntactical errors in the text that need to be corrected.

Author Response

Response to Reviewer 2 Comments

Dear Reviewer 2.

 We sincerely responded to the opinions you gave us and implemented a revision of the manuscipt.

Thanks to your meaningful comments, we were able to improve the quality of this manuscript. Thanks again for the review.

Sincerely,

From Corresponding author.

Point 1: ABSTRACT: Abbreviations are discouraged in this section. Please remove them.

Response 1: According to your opinion, all abbreviations presented in the abstract have been removed. thank you

Point 2: INTRODUCTION: Although this is a fairly extensive section, the small number of bibliographical references used by the Authors to set out the theoretical background of the subject matter is striking. There is a lack of recent and/or relevant references on the object of study of postural control and its different modes of evaluation (doi: 10.3390/su12031222) or on the different applications of artificial intelligence in Health Sciences.

Response 2: We considered the references you recommended and supplemented the introduction. Thanks for the good example material.

Point 3: METHODS: This section should begin by defining the research methodology that has been applied. In this regard, authors should use the relevant EQUATOR Network checklist to ensure that they provide all the necessary information (and in the right order) throughout the manuscript (but especially in this section to facilitate the reproducibility of the research).

Response 3: I checked the EQUATOR Network checklist you informed me. Prior to that, the author guide of this journal "bioengineering" and the various reference manuscripts published in this journal were checked and organized in order. Thank you for your kind comments.

Point 4: RESULTS: Tables should make all abbreviations explicit at the bottom. Descriptive results such as mean or standard deviation should include only one decimal place.

Response 4: Based on your comments, we have added explanations for all abbreviations at the bottom of each table. In addition, it is appropriate to place one decimal for the mean and standard error of the table, but since the values ​​have very small values, it is appropriate to place at least two decimals, so this is expressed. Thank you for your understanding. thank you

Point 5: DISCUSSION: This section suffers from the same shortcomings as the Introduction in relation to the limited use of bibliographical references to support the Authors' arguments. Right now, many sentences, lacking references, seem like speculation

Response 5: I think it is correct to explain how the Stability Index prediction algorithm was derived and applied using the CoP variables in more detail in the method “2.2. Development and validation of balance ability model”. It can be confirmed with the supplementary information in the Development and validation of balance ability model section. Please understand that the results of CTSIB are not presented in figure form because each variable is derived as a non-linear value.

Point 6: CONCLUSIONS: The clinical impact of the research is not adequately reflected in this section.

Response 6: The balance ability estimation model and data acquisition results similarity were confirmed that between the WBB and flexible force mat. The Stability Index prediction can be used instead of expensive commercialized systems, and a model that can predict CoP variables that affect balance ability was identified and developed. This model is more meaningful because it is based on measurements of real elderly people. This clinical impact was added to the conclusion. thank you.

Reviewer 3 Report

This article presents a machine learning-based approach for the diagnostic of elderly balance ability using a stability index estimation algorithm. A piezo-resistive force mat was fabricated, and a Stability Index prediction algorithm was developed and applied using the center of pressure (CoP) variables. The overall paper is interesting and relevant, however, some comments are given to the authors to improve the manuscript as follows:

1- The authors have used four very basic machine learning tools which are linear regression, support vector machine (SVM) regression, generalized additive model (GAM) regression, and regression tree. The training and testing procedure is not clear. How the hyperparameters of each model were selected? Regardless the R-score which might not give a full picture of the efficiency of the models, what about the training and testing statistical analysis? Please discuss!

2- A regression equation was developed in order to model the experimental data. It is not clear to me how the equation was written based on the four models. Please discuss in detail!

3- The machine learning models are not very significant, some other models can be tried which surely will provide better results such as the ANNs, Random Forest models, Gaussian regression, etc. I suggest the use of more reliable machine learning tools or at least consider this issue as future work.

4- On what basis the input variables of the models were selected? How the particular frequencies or other types of dynamic properties might affect the results? Please discuss!

Author Response

Response to Reviewer 3 Comments

Dear Reviewer 3

 We sincerely responded to the opinions you gave us and implemented a revision of the manuscipt.

Thanks to your meaningful comments, we were able to improve the quality of this manuscript. Thanks again for the review.

Sincerely,

From Corresponding author.

Point 1: 1- The authors have used four very basic machine learning tools which are linear regression, support vector machine (SVM) regression, generalized additive model (GAM) regression, and regression tree. The training and testing procedure is not clear. How the hyperparameters of each model were selected? Regardless the R-score which might not give a full picture of the efficiency of the models, what about the training and testing statistical analysis? Please discuss!

Response 1: According to your opinion, since the training and test procedures are based on the general model development method, we have added more specific information about the actual configuration of the training-set and test-set, and the procedure for using the data-set. The hyperparameter setting method for each model is described in “2.2. In the “Development and validation of balance ability model” section, explanations have been added (line171-176). thank you.

Point 2: A regression equation was developed in order to model the experimental data. It is not clear to me how the equation was written based on the four models. Please discuss in detail!

Response 2: “Figure 5” means the formula of the finally selected regression model. Formulas for regression models of all methods were not presented due to limitations in the amount of paper writing. Thank you for your understanding.

Point 3: The machine learning models are not very significant, some other models can be tried which surely will provide better results such as the ANNs, Random Forest models, Gaussian regression, etc. I suggest the use of more reliable machine learning tools or at least consider this issue as future work.

Response 3: We agree with your opinion. This study developed a model using very basic machine learning techniques. The reason for this is that it is easy to derive a model that can be explained by formulas and update it with software code, so this method was chosen. However, in order to improve the accuracy of the model, we are considering developing a deep learning-based unsupervised learning model such as the DNN you recommended, and it is actually in progress. The results of this will be presented in a later paper work. The fact that deep learning techniques were not utilized was added to the limitations of this study (line 338-340). thank you for your kind comments.

Point 4: On what basis the input variables of the models were selected? How the particular frequencies or other types of dynamic properties might affect the results? Please discuss!

Response 4: The input variables used in the use of the model are general variables that have already been used for evaluation of balance ability in various preceding studies. However, this study is characterized by developing a model to predict a specific value called “stability index”. In addition, particular frequencies or other types of dynamic properties do not have a large effect because filtering is performed by digital signal processing on the measured data. This is secured through various preceding studies. thank you

Round 2

Reviewer 1 Report

thanks, no further comments

Reviewer 3 Report

The paper can be accepted for publication.